# Constraint-Conditioned Policy Optimization for Versatile Safe Reinforcement Learning

**Yihang Yao**[*1], **Zuxin Liu**[*1], **Zhepeng Cen**[1], **Jiacheng Zhu**[1,3],
**Wenhao Yu**[2], **Tingnan Zhang**[2], **Ding Zhao**[1]
[1] Carnegie Mellon University, [2] Google DeepMind, [3] Massachusetts Institute of Technology
[*] Equal contribution, {yihangya, zuxinl}@andrew.cmu.edu

## Abstract

Safe reinforcement learning (RL) focuses on training reward-maximizing agents subject to pre-defined safety constraints. Yet, learning versatile safe policies that can adapt to varying safety constraint requirements during deployment without retraining remains a largely unexplored and challenging area. In this work, we formulate the versatile safe RL problem and consider two primary requirements: training efficiency and zero-shot adaptation capability. To address them, we introduce the Constraint-Conditioned Policy Optimization (CCPO) framework, consisting of two key modules: (1) Versatile Value Estimation (VVE) for approximating value functions under unseen threshold conditions, and (2) Conditioned Variational Inference (CVI) for encoding arbitrary constraint thresholds during policy optimization. Our extensive experiments demonstrate that CCPO outperforms the baselines in terms of safety and task performance, while preserving zero-shot adaptation capabilities to different constraint thresholds data-efficiently. This makes our approach suitable for real-world dynamic applications.

## 1 Introduction

Safe reinforcement learning (RL) has emerged as a promising approach to address the challenges faced by agents operating in complex, real-world environments [1], such as autonomous driving [2], home service [3], and UAV locomotion [4]. Safe RL aims to learn a reward-maximizing policy within a constrained policy set [5–10]. By explicitly accounting for safety constraints during policy learning, agents can better reason about the trade-off between task performance and safety constraints, making them well-suited for safety-critic applications [11].

Despite the advances in safe RL, the development of a versatile policy that can adapt to varying safety constraint requirements during deployment without retraining remains a largely unexplored area. Investigating versatile safe RL is crucial due to the inherent trade-off between task reward and safety requirement [12, 13]: stricter constraints typically lead to more conservative behavior and lower task rewards. For example, an autonomous vehicle can adapt to different thresholds for driving on an empty highway and crowded urban area to maximize transportation efficiency. Consequently, learning a versatile policy allows agents to efficiently adapt to diverse constraint conditions, enhancing their applicability and effectiveness in real-world scenarios [14].

This paper studies the problem of training a versatile safe RL policy capable of adapting to tasks with different cost thresholds. The primary challenges are two-fold:

**(1) Training efficiency.** A straightforward approach is to train multiple policies under different constraint thresholds, then the agent can switch between policies for different safety requirements. However, this method is sampling inefficient, making it unsuitable for most practical applications, as the agent may only collect data under a limited number of thresholds during training.

37th Conference on Neural Information Processing Systems (NeurIPS 2023).

**(2) Zero-shot adaptation capability.** Constrained optimization-based safe RL approaches rely on fixed thresholds during training [15], while recovery-based safe RL methods require a pre-defined backup policy to correct agent's unsafe behaviors [5–7]. Therefore, current safe RL training paradigms face challenges in adapting the learned policy to accommodate unseen safety thresholds.

To tackle the challenges outlined above, we introduce the Conditioned Constrained Policy Optimization (CCPO) framework, a sampling-efficient algorithm for versatile safe reinforcement learning that achieves zero-shot generalization to unseen cost thresholds during deployment. Our method consists of two integrated components: Versatile Value Estimation (VVE) and Conditioned Variational Inference (CVI). The first VVE module is inspired by transfer learning in RL [16, 17], which utilizes value function representation learning to estimate value functions for the versatile policy under unseen threshold conditions. The second CVI module aims to encode arbitrary threshold conditions during policy training. Our key contributions are summarized as follows:

**1. We frame safe RL beyond pre-defined constraint thresholds as a versatile learning problem.** This perspective highlights the limitations of most existing constrained-optimization-based approaches and motivates the development of CCPO based on conditional variational inference. Importantly, CCPO can generalize to diverse unseen constraint thresholds without retraining the policy.

**2. We introduce two key techniques, VVE and CVI, for safe and versatile policy learning.** To the best of our knowledge, our method is the first successful online safe RL approach capable of achieving zero-shot adaptation for unseen thresholds while preserving safety. Our theoretical analysis further provides insights into our approach's data efficiency and safety guarantees.

**3. We conduct comprehensive evaluations of our method on various safe RL tasks.** The results demonstrate that CCPO outperforms baseline methods in terms of both safety and task performance for varying constraint conditions. The performance gap is notably larger in tasks with the high-dimensional state and action space, wherein all baseline methods fail to realize safe adaptation.

## 2 Related Work

**Safe RL** has been approached through various methods. Some techniques leverage domain knowledge of the target problem to enhance the safety of an RL agent [18–26]. Another line of work employs constrained optimization techniques to learn a constraint-satisfaction policy [27, 1, 28], such as the Lagrangian-based approach [29–31], where the Lagrange multipliers can be optimized via gradient descent along with the policy parameters [32, 33, 12]. Alternatively, other works approximate the non-convex constrained optimization problem with low-order Taylor expansions [15] or through variational inference [34], then solve for the dual variable using convex optimization [35–38]. However, most existing approaches consider a fixed constraint threshold during training, which can hardly be deployed for different threshold conditions after training.

**Transfer learning in RL.** The concept of transfer learning, also recognized as knowledge transfer, denotes a sophisticated technique that exploits external knowledge harnessed from various domains to enhance the learning trajectory of a specified target task [39]. Transfer learning in RL can be categorized from multiple perspectives, such as skill composition for novel tasks [40–43], parameter transfer [44], and feature representation transfer [45–47]. Among these, the methodologies leveraging Successor Features (SFs) [48, 17] are particularly relevant to this work. These methodologies operate under the assumption that the reward function can be deconstructed into a linear combination of features, and they further extend the successor representation to decouple environmental dynamics from rewards [16, 49]. However, most existing works using SFs consider transfer learning problems among tasks with different reward functions but not with different task conditions.

## 3 Problem Formulation

### 3.1 Safe RL with Constrained Markov Decision Process

Constrained Markov Decision Process (CMDP) $\mathcal{M}$ is defined by the tuple $(\mathcal{S}, \mathcal{A}, \mathcal{P}, r, c, \mu_0)$ [50], where $\mathcal{S}$ is the state space, $\mathcal{A}$ is the action space, $\mathcal{P} : \mathcal{S} \times \mathcal{A} \times \mathcal{S} \to [0, 1]$ is the transition function, $r : \mathcal{S} \times \mathcal{A} \times \mathcal{S} \to \mathbb{R}$ is the reward function, and $\mu_0 : \mathcal{S} \to [0, 1]$ is the initial state distribution. CMDP augments MDP with an additional element $c : \mathcal{S} \times \mathcal{A} \times \mathcal{S} \to \mathbb{R}^+$ to characterize the cost of violating the constraint. Note that in this work we use a single constraint for ease of demonstration.

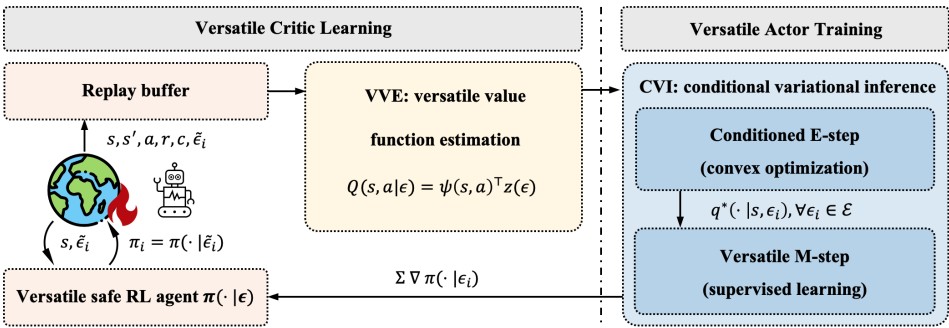

Figure 1: Proposed framework

A safe RL problem is specified by a CMDP and a constraint threshold $\epsilon \to [0, +\infty)$. Let $\pi : \mathcal{S} \times \mathcal{A} \to [0, 1]$ denote the policy and $\tau = \{s_1, a_1, ...\}$ denote the trajectory. We use shorthand $\mathbf{f}_t = \mathbf{f}(s_t, a_t, s_{t+1}), \mathbf{f} \in \{r, c\}$ for simplicity. The value function is $V_{\mathbf{f}}^{\pi}(\mu_0) = \mathbb{E}_{\tau \sim \pi, s_0 \sim \mu_0}[\sum_{t=0}^{\infty} \gamma^t \mathbf{f}_t], \mathbf{f} \in \{r, c\}$, which is the expectation of discounted return under the policy $\pi$ and the initial state distribution $\mu_0$. The goal of safe RL is to find the policy that maximizes the reward return while limiting the cost return under the pre-defined threshold $\epsilon$:

$$\pi^* = \arg\max_{\pi} V_r^{\pi}(\mu_0), \quad s.t. \quad V_c^{\pi}(\mu_0) \le \epsilon. \tag{1}$$

### 3.2 Versatile Safe RL beyond a Single Threshold

We consider the versatile safe RL problem beyond a single pre-defined constraint threshold. Specifically, we consider a set of thresholds $\epsilon \in \mathcal{E}$ and a constraint-conditioned policy space: $\pi(\cdot|\epsilon) : \mathcal{S} \times \mathcal{A} \times \mathcal{E} \to [0, 1]$. We can then formulate the versatile safe RL problem as finding the optimal versatile policy $\pi^*(\cdot|\epsilon)$ that maximizes the reward while sticking to the corresponding threshold condition on a range of constraint thresholds $\epsilon \in \mathcal{E}$, i.e.,

$$\pi^*(\cdot|\epsilon) = \arg\max_{\pi} V_r^{\pi}(\mu_0), \quad s.t. \quad V_c^{\pi}(\mu_0) \le \epsilon, \quad \forall \epsilon \in \mathcal{E}. \tag{2}$$

The generated action is subsequently based on the state and threshold: $a \sim \pi(s|\epsilon)$. A successful versatile training algorithm should satisfy as least two core properties:

**(1) Thresholds Sampling Efficiency:** The training dataset $D = \bigcup_{i=1}^{N} D_i$ is collected through a limited set of thresholds $D_i \sim \pi(\cdot|\tilde{\epsilon}_i), \forall \epsilon_i \in \tilde{\mathcal{E}}$, with $\tilde{\mathcal{E}} \subset \mathcal{E}, |\tilde{\mathcal{E}}| = N$, where $N$ denotes the number of behavior policies with pre-specified constraint conditions during training.

**(2) Safety for Varying Thresholds:** Given that the agent is trained under a restricted range of threshold conditions, ensuring safety when adapting to unseen thresholds is of significant importance.

We identify two key challenges to meet these requirements: (1) Q function estimation for unseen threshold conditions with limited behavior policy data and (2) encoding arbitrary safety constraint conditions in the versatile policy training. To address these challenges, we propose the Constraint-Conditioned Policy Optimization (CCPO) method as follows.

## 4 Method

As illustrated in Figure. 1, CCPO is comprised of two parts. The first **versatile critic learning** part involves data collection via several behavior agents, each under their respective target constraint thresholds. The goal is to learn feature representations for both the state-action pair feature and the target thresholds, facilitating linear disentanglement of Q values and hence enabling generalization to unseen thresholds. This step is inspired by the concept of successor feature in RL transfer learning [16], and we term it Versatile Value Estimation (VVE).

In the second **versatile agent training** part, we train the policy to be responsive to a range of unseen thresholds based on the well-trained value functions. Our key insight is to adopt the variational inference safe RL framework [34], which allows us to compute the optimal non-parametric policy

distributions for various thresholds through convex optimization, and, subsequently, use them to train the constraint-conditioned policy via supervised learning. With this Conditional Variational Inference (CVI) step, the policy can achieve zero-shot adaptation to unseen thresholds without the necessity for behavior agents to collect data under corresponding conditions. We introduce each step as follows.

## 4.1 Versatile value estimation

It has been shown that inherent trade-offs exist between the reward and cost values for the optimal policy under varying cost thresholds [13]. Typically, stricter safety requirements tend to be accompanied by reduced rewards. Hence, the estimation of Q-value functions becomes increasingly crucial when dealing with unseen thresholds that are not encountered by the behavior agents. To this end, we propose the versatile critics learning module, which disentangles observations and target thresholds within a latent feature space. The assumption regarding the decomposition is as follows.

**Assumption 1** (Critics linear decomposition). *The versatile Q functions $Q_{\mathbf{f}}^*$ with respect to the optimal versatile policy $\pi^*$ can be represented as:*

$$Q_{\mathbf{f}}^*(s, a|\epsilon) = \boldsymbol{\psi}_{\mathbf{f}}(s, a)^\top \boldsymbol{z}_{\mathbf{f}}^*(\epsilon), \quad \mathbf{f} \in \{r, c\}, \tag{3}$$

*where $||\boldsymbol{\psi}_{\mathbf{f}}(s, a)||_\infty \leq K_{\mathbf{f}}$ is the feature for the function of the state-action pair $(s, a)$, $K_{\mathbf{f}}$ are constants, and $\boldsymbol{z}_{\mathbf{f}}^*$ is the optimal constraint-conditioned policy feature, which only depends on the policy condition $\epsilon$ for a specified task. The dimension of $\boldsymbol{\psi}_{\mathbf{f}}(s, a)$ and $\boldsymbol{z}_{\mathbf{f}}^*(\epsilon)$ are both $M$.*

Note that Assumption 1 is reasonable and widely accepted in the RL transfer learning literature with successor features [17], as it is reasonable to find a high-dimensional feature space to decompose the Q functions into the product of feature functions $\psi_{\mathbf{f}}(s, a)$ and the latent vectors $\boldsymbol{z}_{\mathbf{f}}(\epsilon)$ [51]. As shown in Theorem 1, by learning $\boldsymbol{\psi}_{\mathbf{f}}(s, a)$ and $\boldsymbol{z}_{\mathbf{f}}(\epsilon)$ jointly and adding norm constraints on the feature function $||\boldsymbol{\psi}_{\mathbf{f}}(s, a)||_\infty \leq K_{\mathbf{f}}$, we can efficiently encode the threshold information $\epsilon$ into the Q functions and achieve accurate estimations for unseen thresholds. This is the basis for our method's data-efficient training. To further facilitate theoretical analysis, we assume that the optimal constraint-conditioned policy feature $\boldsymbol{z}_{\mathbf{f}}^*(\epsilon)$ can be approximated by polynomial functions:

**Assumption 2** (Polynomial feature space). *The optimal constraint-conditioned policy feature $\boldsymbol{z}_{\mathbf{f}}^*$ can be approximated by $\boldsymbol{z}_{\mathbf{f}}^*(\epsilon) = Poly(\epsilon, p) + \boldsymbol{e}$, meaning each element of $\boldsymbol{z}_{\mathbf{f}}^*$ is a $p$-degree polynomial of $\epsilon$, and $\boldsymbol{e}$ is the remainder. Each component for $\boldsymbol{e}$ follows $e_j \sim \mathcal{N}(0, \sigma_j^2), j = i, ..., M$, and denote $\sigma = \max_j \sigma_j$.*

Note that the degree $p$ corresponds to the $\boldsymbol{z}(\epsilon)$ model representation capability. Based on the above assumptions, we can derive the Q function estimation error bound as follows.

**Theorem 1** (Bounded estimation error). *Denote $\epsilon_L$ and $\epsilon_H$ are the lower and upper bound of the target threshold interval for $\mathcal{E}$. Suppose the threshold conditions $\{\tilde{\epsilon}_i\}_{i=1,2,...,N}$ for behavior policies are selected to divide the interval $[\epsilon_L, \epsilon_H]$ evenly, then with confidence level $1 - \alpha$, the estimation error of versatile Q functions conditioned on arbitrary $\epsilon \in [\epsilon_L, \epsilon_H]$ can be bounded by:*

$$||\hat{Q}_{\mathbf{f}}(s, a|\epsilon) - Q_{\mathbf{f}}^*(s, a|\epsilon)|| \leq \frac{z_{\alpha/2} B(p)}{N^{\beta(p)}} \sqrt{\sigma^2 K_{\mathbf{f}}^2 M}, \tag{4}$$

where $B(p)$ and $\beta(p)$ are both functions of the polynomial degree $p$, and $z_{\alpha/2}$ is the upper alpha quantile for the standard Gaussian distribution. The proof and detailed discussion of Theorem 1 and functions $B(p), \beta(p)$ are shown in Appendix B.1. It is worth noting that we normalize the threshold conditions $\epsilon \in [\epsilon_L, \epsilon_H]$ to the interval $[0, 1]$ by $\epsilon = (\epsilon - \epsilon_L)/\epsilon_H$ for numerical stability. Theorem 1 establishes that by decomposing the Q function into the product of $\boldsymbol{\psi}(s, a)$ and $\boldsymbol{z}(\epsilon)$ and jointly learning these components, we can guarantee a bounded estimation error for Q functions under unseen threshold conditions. Furthermore, we can derive the bounded cost violation for unseen thresholds and $\epsilon$-sample complexity analysis based on the theorem, both of which are discussed in proposition 1 and remark 1. We also provide empirical verification of Q function estimation in Appendix D.1.

## 4.2 Conditioned variational inference

Given well-trained versatile Q functions in the previous VVE module, we aim to encode arbitrary threshold constraints during policy learning in the versatile policy learning part. We utilize the *safe*

*RL as inference* framework to achieve this goal, as it decomposes safe RL to a convex optimization followed by supervised learning, both stages readily accommodating varying target thresholds. In contrast to the classical view of safe RL aiming to find the most-rewarding actions while satisfying the constraints, the probabilistic inference perspective finds the feasible (constraint-satisfying) actions most likely to have been taken given future success in maximizing task rewards [34].

Following the RL as inference literature [52, 53], we consider an infinite discounted reward formulation. In the condition that the constraint threshold is $\epsilon_i$, we denote $O = O(s, a)$ as the optimality variable of a state-action pair $(s, a)$, which indicates the reward-maximizing (optimal) event by choosing an action $a$ at a state $s$. Then for a given trajectory $\tau$, the likelihood of being optimal is proportional to the exponential of the discounted cumulative reward: $p(O = 1|\tau) \propto \exp(\sum_t \gamma^t r_t / \alpha)$, where $\alpha$ is a temperature parameter. Since the probability of getting a trajectory $\tau$ under the conditioned policy $\pi(\cdot|\epsilon_i)$ can be expressed as $p_{\pi(\cdot|\epsilon_i)}(\tau) = p(s_0) \prod_{t \geq 0} p(s_{t+1}|s_t, a_t)\pi(a_t|s_t, \epsilon_i)$, the lower bound for the log-likelihood of optimality given the conditioned policy $\pi(\cdot|\epsilon_i)$ is:

$$
\begin{aligned}
\log p_{\pi(\cdot|\epsilon_i)}(O = 1) &= \log \mathbb{E}_{\tau \sim q(\cdot|\epsilon_i)} \frac{p(O = 1|\tau)p_\pi(\tau|\epsilon_i)}{q(\tau|\epsilon_i)} \\
&\geq \mathbb{E}_{\tau \sim q(\cdot|\epsilon_i)} \log \frac{p(O = 1|\tau)p_{\pi(\cdot|\epsilon_i)}(\tau)}{q(\tau|\epsilon_i)} \\
&\propto \mathbb{E}_{\tau \sim q(\cdot|\epsilon_i)} [\sum_{t=0}^{\infty} \gamma^t r_t] - \alpha D_{\mathrm{KL}}(q(\tau|\epsilon_i)\|p_{\pi(\cdot|\epsilon_i)}(\tau)) := \mathcal{J}(q, \pi|\epsilon_i),
\end{aligned}
\tag{5}
$$

where the inequality follows Jensen's inequality, and $q(\tau|\epsilon_i)$ is an auxiliary trajectory-wise variational distribution conditioned on $\epsilon_i$. $\mathcal{J}(q, \pi|\epsilon_i)$ in equation (5) is the evidence lower bound (ELBO) to reach the reward optimality under condition $\epsilon_i$. Since $q(\tau|\epsilon_i) = p(s_0) \prod_{t \geq 0} p(s_{t+1}|s_t, a_t)q(a_t|s_t, \epsilon_i)$, we have the following ELBO over the state and constraint conditioned action distribution $q(a|s, \epsilon_i)$:

$$
\mathcal{J}(q, \theta|\epsilon_i) = \mathbb{E}_{\rho_q(\cdot|\epsilon_i)} \Big[ \sum_{t=0}^{\infty} \gamma^t r_t - \alpha D_{\mathrm{KL}}(q(\cdot|\epsilon_i)\|\pi_\theta(\cdot|\epsilon_i)) \Big] + \log p(\theta)
\tag{6}
$$

where $\rho_q(s|\epsilon_i)$ is the stationary state distribution induced by $q(\cdot|s, \epsilon_i)$ and $\rho_0$, $\theta$ refers to the parameters for policy $\pi$, and $p(\theta)$ is a prior distribution over the parameters. Note we overload $q$ by using it both in $q(a|s, \epsilon_i)$ and $q(\tau|\epsilon_i)$.

We utilize the Expectation-Maximization (EM)-based RL algorithms, which alternate to improve $\mathcal{J}(q, \pi|\epsilon_i)$ in terms of $q(\tau|\epsilon_i)$ and $p_{\pi(\cdot|\epsilon_i)}(\tau)$ to improve the likelihood of optimality [54, 55, 34]. Denote the feasible distribution family for the conditioned variational distribution $q(\cdot|s, \epsilon_i)$ as:

$$
\Pi_{\mathcal{Q}}^{\epsilon_i} := \{q(a|s, \epsilon_i) : \mathbb{E}_{\tau \sim q(\cdot|\epsilon_i)}[\sum_{t=0}^{\infty} \gamma^t c_t] \leq \epsilon_i, a \in \mathcal{A}, s \in \mathcal{S}\},
\tag{7}
$$

which is a set of all the state-conditioned action distributions that satisfy the safety constraint specified by $\epsilon_i$. Then the E-step optimizes $q(\tau|\epsilon_i)$ to maximize the reward return within the trust region of the old policy and within $\Pi_{\mathcal{Q}}^{\epsilon_i}$, while the M-step aims to minimize the KL divergence between $p_{\pi(\cdot|\epsilon_i)}(\tau)$ and $q(\tau|\epsilon_i)$ by updating the parametrized policy in a supervised learning fashion. Since Off-policy deep RL techniques can be used during training, the EM updating steps are more data efficient.

The key strength of using the variational inference framework lies in its ability to encode arbitrary threshold conditions during policy learning, as shown in (7), a feat that is challenging for other methods, such as those based on primal-dual algorithms. Optimizing the factorized lower bound $\mathcal{J}(q, \theta|\epsilon_i)$ w.r.t $q(\cdot|\epsilon_i)$ within the feasible distribution family and the policy parameter $\theta$ iteratively with one single threshold $\epsilon_i$ via EM yields the CVPO method [34], which is the basis of our method. We introduce the modified constraint-conditioned E-step and M-step as follows.

**Constraint-Conditioned E-step:** The conditioned E-step aims to find the optimal variational distribution $q(\cdot|\epsilon_i) \in \Pi_{\mathcal{Q}}^{\epsilon_i}$ that maximizes the reward return while satisfying the safety condition defined by $\epsilon_i$. At the $j$-$th$ iteration, We can write the ELBO objective w.r.t $q$ as a constrained

optimization problem (see Appendix B for proofs):

$$\max_{q(a|s,\epsilon_i)} \mathbb{E}_{\rho_q} \left[ \int q\left(a|s,\epsilon_i\right) \hat{Q}_r^{\pi_{\theta_j}}\left(s,a|\epsilon_i\right) da \right]$$

$$\text{s.t. } \mathbb{E}_{\rho_q} \left[ \int q\left(a|s,\epsilon_i\right) \hat{Q}_c^{\pi_{\theta_j}}\left(s,a|\epsilon_i\right) da \right] \le \epsilon_i, \tag{8}$$

$$\mathbb{E}_{\rho_q} \left[ D_{\mathrm{KL}}\left( q\left(a|s,\epsilon_i\right) \| \pi_{\theta_j}\left(\cdot|\epsilon_i\right) \right) \right] \le \kappa;$$

where $\hat{Q}_{\mathbf{f}}(\cdot|\epsilon_i)$ is the versatile Q functions as introduced in section 4.1, the first inequality constraint represents the constraint defined in (7) and the last term in the constraint is the trust region with the old policy defined by KL distance $\kappa$. Inspired by [54], we use the solution of the optimal variational distribution $q_i^* = q_i^*(a|s,\epsilon_i)$ for arbitrary safety constraint $\epsilon_i$, which has the closed form:

$$q_i^* = \frac{\pi_{\theta_j}(\cdot|\epsilon_i)}{Z(s,\epsilon_i)} \exp\left( \frac{\hat{Q}_r^{\pi_{\theta_j}}(\cdot|\epsilon_i) - \lambda_i^* \hat{Q}_c^{\pi_{\theta_j}}(\cdot|\epsilon_i)}{\eta_i^*} \right), \tag{9}$$

where $Z(s,\epsilon_i)$ is a normalizer to make sure $q_i^*$ is a valid distribution, and the dual variables $\eta_i^*$ and $\lambda_i^*$ are the solutions of the following convex optimization problem (see Appendix B for details):

$$\min_{\lambda_i,\eta_i \ge 0} g(\eta_i,\lambda_i) = \lambda_i \epsilon_i + \eta_i \kappa \mathbb{E}_{\rho_q} \left[ \log \mathbb{E}_{\pi(\cdot|\epsilon_i)} \left[ \exp\left( \frac{\hat{Q}_r(\cdot|\epsilon_i) - \lambda_i \hat{Q}_c(\cdot|\epsilon_i)}{\eta_i} \right) \right] \right]. \tag{10}$$

Then we can encode arbitrary safety constraints by calculating the optimal distribution $q_i^*(a|s,\epsilon_i)$ regarding the corresponding condition $\epsilon_i$ efficiently with (9). The term $q_i^*(a|s,\epsilon_i)$ means when conditioned on $\epsilon_i$, the probability of taking $a$ at $s$ for the optimal feasible policy.

**Versatile M-step:** After the constraint-conditioned E-step, we obtain a set of optimal feasible variational distribution $q_i^* = q_i^*(\cdot|s,\epsilon_i)$ for each constraint threshold $\epsilon_i$. In the versatile M-step, we aim to improve the ELBO (5) w.r.t the policy parameter $\theta$ for $\epsilon_i \in \mathcal{E}$.

$$\mathcal{J}(\theta|\epsilon_i) = \mathbb{E}_{\rho_q} \left[ \alpha \mathbb{E}_{q_i^*} \left[ \log \pi_\theta(a|s,\epsilon_i) \right] \right] + \log(p|\epsilon_i) \tag{11}$$

Using a Gaussian prior for each threshold-conditioned policy, this problem can be further converted to the following supervised-learning problem with KL-divergence constraints [54, 55]:

$$\max_\theta \mathbb{E}_{\rho_q} \left[ \sum_{i=1}^{|\mathcal{E}|} \mathbb{E}_{q_i^*} \left[ \log \pi_\theta(a|s,\epsilon_i) \right] / |\mathcal{E}| \right] \ s.t. \ \mathbb{E}_{\rho_q} \left[ D_{\mathrm{KL}}(\pi_{\theta_j}(a|s,\epsilon_i) \| \pi_\theta(a|s,\epsilon_i)) \right] \le \gamma \ \forall i, \tag{12}$$

where $\mathcal{E}$ is the set for all the sampled versatile policy conditions $\{\epsilon_i\}$ in fine-tuning stage of training. The constraint in (12) is a regularizer to stabilize the policy update. The proposed CCPO for versatile safe RL and implementation details are summarized in Appendix C.

### 4.3 Theoretical analysis

**Proposition 1** (Bounded safety violation). *With the threshold conditions $\tilde{\epsilon}_i \in \tilde{\mathcal{E}}$ for behavior policies selected to divide the target condition interval $[\epsilon_L, \epsilon_H]$ evenly, and with confidence level $1 - \alpha$, the cost violation of versatile policy under arbitrary threshold condition $\epsilon \in [\epsilon_L, \epsilon_H]$ is bounded as:*

$$V_c^{\pi(\mu_0|\epsilon)} - \epsilon \le \frac{z_{\alpha/2} B(p)}{N^{\beta(p)}} \sqrt{\sigma^2 K_c^2 M}, \tag{13}$$

The proof is shown in Appendix B.1. Proposition 1 ensures that the cost violation of the versatile safe RL agent on unseen thresholds can be bounded if the selected behavior policy conditions divide the interval $[\epsilon_L, \epsilon_H]$ evenly. We can observe that the bound (13) is proportional to $\sqrt{K_c^2 M}$. Since larger $K_c$ and $M$ correspond to a wider range of threshold conditions, this safety violation bound is related to the interval range. Also, we provide the complexity analysis for the $\epsilon$-sample, i.e., the estimation error corresponding to the number of behavior policies $N$ as shown in remark 1.

**Remark 1** ($\epsilon$-sample complexity analysis). *The estimation error for Q functions and safety violation bound decreases as the number of behavior policies $N$ increases. The decreasing rate is proportional to $\frac{1}{N^{\beta(p)}}$, where the exponent of $N$ is related to $p$, which is the representation capabilities of the model to represent the constraint-conditioned policy feature $z(\epsilon)$. When $p$ increases, $\beta$ also increases as shown in the Appendix B.1, which means when the model capability is high, the proposed method significantly becomes more data-efficient.*

The functions $\beta(p), B(p)$ are shown in Appendix B.1. Since CCPO is under the "RL as inference" framework, we also enjoy many benefits as revealed in previous works, such as the optimality guarantees and training robustness [34], see Appendix A for details.

## 5 Experiment

We aim to answer five major questions in the experiment section: (1) Can we achieve versatile safe RL by simply applying a linear combination of single-threshold policies (2) Can we combine safe RL algorithms with a constraint-conditioned actor to achieve versatile safe RL? (3) What is the performance of our proposed CCPO method in versatile safe RL tasks in terms of constraint violation and reward? (4) How $\epsilon$-efficient CCPO is compared to exhaustively training the safe RL agents? (5) What is the contribution of each component in CCPO contribute to the overall performance? We adopt the following experiment setting to address these questions.

**Task.** The simulation environments are from a publicly available benchmark [56]. We consider two tasks (Run and Circle) and four robots (Ball, Car, Drone, and Ant) which have been used in many previous works as the testing ground [13–15]. For the Run task, the agents are rewarded for running fast between two boundaries and are given constraint violation cost if they run across the boundaries or exceed an agent-specific velocity threshold. For the Circle task, the agents are rewarded for running in a circle but are constrained within a safe region smaller than the target circle's radius. We name the tasks as `Ball-Circle`, `Car-Circle`, `Drone-Circle`, `Drone-Run`, and `Ant-Run`.

**Constraint-conditioned Baselines.** We design these baselines by directly integrating the threshold as a part of the state in the CMDP tuple, $\bar{s} = [s; \epsilon]$. The policy is optimized with behavior policy conditions only. We adopt commonly used off-policy safe RL algorithms, `SAC-Lag` and `DDPG-Lag`, and name the proposed baselines as `V-SAC-Lag` and `V-DDPG-Lag`.

**Policy linear combination baselines.** We also compare our method with single-threshold policy combinations. Denote $\epsilon$ as an unseen target threshold for adaptation, and $\epsilon_1, \epsilon_2$ as two behavior policy conditions closest to $\epsilon$. Then the policy for $\epsilon$ is the combination of $\pi(\cdot|\epsilon_1), \pi(\cdot|\epsilon_2)$:

$$\pi(\cdot|\epsilon) = w_1\pi(\cdot|\epsilon_1) + w_2\pi(\cdot|\epsilon_2); \quad w_1 = (\epsilon_2 - \epsilon)/(\epsilon_2 - \epsilon_1), \ w_2 = (\epsilon - \epsilon_1)/(\epsilon_2 - \epsilon_1) \quad (14)$$

This method is designed for both threshold interpolation and extrapolation, i.e., the coefficients $w_1$ or $w_2$ can be negative. This baseline draws inspiration from the safe control theory, which suggests the safe input component is proportional to the conservativeness level [57]. To this end, we use two strong on-policy methods `PPO-Lag` and `TRPO-Lag` to train the single-threshold agents and named the corresponding baselines as `C-PPO-Lag` and `C-TRPO-Lag`. More baselines and results can be found in the Appendix.

**Metrics:** We compare the methods in terms of episodic reward (the higher, the better) and episodic constraint violation cost (the lower, the better) on each evaluated threshold condition, which have been used in many related works [14, 58]. For all the results shown in section 5.1 and 5.3, the behavior policy conditions are $\tilde{\mathcal{E}} = \{20, 40, 60\}$ and the threshold conditions for evaluation are set to be $\{10, 15, ..., 70\}$. We take the average of the episodic reward (Avg. R) and constraint violation (Avg. CV) as the main comparison metrics. The constraint violation for threshold $\epsilon$ is defined as:

$$CV = \max\{0, \Sigma_\tau c_t - \epsilon\} \quad (15)$$

As mentioned in previous works [58], the general evaluation criteria in safe RL are: (1) method A is better than method B if A achieves better safety performance than B. (2) If both A and B satisfy constraints, the one with the higher reward is better. We also report the average performance solely on unseen thresholds (Avg. R-G and Avg. CV-G) to characterize the adaptation capability. More experiments can be found in Appendix D.

### 5.1 Main Results and Analysis

The evaluation results are shown in Figure. 2 and Table 1. We shade the two safest agents with the lowest averaged cost violation values.

First, we can observe that our method outperforms the modified versatile safe RL baselines `V-SAC-Lag` and `V-DDPG-Lag` that directly concatenate the threshold condition into the state. Although `V-SAC-Lag` or `V-DDPG-Lag` gets higher rewards on simple-dynamics tasks `Ball-Circle`,

Table 1: Evaluation results of proposed CCPO method and the proposed versatile safe RL baselines. ↑: the higher reward, the better. ↓: the lower constraint violation (minimal 0), the better. The models are evaluated on a series of threshold conditions and we report the averaged reward and constraint violation values on all evaluation thresholds and generalized thresholds. Each value is reported as mean ± standard deviation for 50 episodes and 5 seeds. We shade the two safest agents with the lowest averaged cost violation values.

| Task | Stats | CCPO (ours) | Constraint-conditioned | | Linear combination | |
| | | | V-SAC-Lag | V-DDPG-Lag | C-PPO-Lag | C-TRPO-Lag |
|---|---|---|---|---|---|---|
| Ball-Circle | Avg. R ↑ | 710.86±20.47 | 774.16±20.34 | 762.61±58.65 | 637.85±14.03 | 699.38±1.94 |
| | Avg. CV ↓ | 0.59±0.31 | 5.32±5.00 | 2.81±1.12 | 3.11±1.64 | 4.50±0.08 |
| | Avg. R-G ↑ | 699.04±20.48 | 766.52±22.59 | 756.67±58.48 | 667.89±12.17 | 699.14±2.05 |
| | Avg. CV-G ↓ | 0.83±0.42 | 6.29±5.72 | 3.53±1.26 | 3.40±1.75 | 5.59±0.25 |
| Car-Circle | Avg. R ↑ | 406.06±6.30 | 331.80±11.57 | 448.82±18.65 | 440.01±2.59 | 461.14±1.39 |
| | Avg. CV ↓ | 1.60±0.91 | 12.18±4.65 | 14.48±8.14 | 9.09±1.52 | 7.84±1.71 |
| | Avg. R-G ↑ | 401.53±5.59 | 331.19±11.00 | 445.32±17.42 | 438.31±3.03 | 460.72±1.15 |
| | Avg. CV-G ↓ | 1.49±0.38 | 12.74±4.32 | 14.63±8.69 | 11.07±1.58 | 9.14±2.01 |
| Drone-Circle | Avg. R ↑ | 630.55±40.03 | 693.69±22.37 | 734.58±49.69 | 392.64±23.13 | 380.77±18.62 |
| | Avg. CV ↓ | 0.32±0.38 | 13.24±8.80 | 19.62±11.15 | 0.45±0.38 | 6.55±1.95 |
| | Avg. R-G ↑ | 625.51±40.12 | 699.14±24.88 | 730.29±48.43 | 342.77±19.06 | 291.87±19.88 |
| | Avg. CV-G ↓ | 0.47±0.55 | 14.97±10.10 | 19.44±10.36 | 0.21±0.09 | 7.23±2.03 |
| Drone-Run | Avg. R ↑ | 458.69±12.98 | 355.61±35.44 | 244.60±48.29 | 398.88±21.53 | 461.70±4.91 |
| | Avg. CV ↓ | 0.23±0.25 | 8.66±4.30 | 11.33±9.63 | 9.46±5.63 | 47.97±3.49 |
| | Avg. R-G ↑ | 455.64±11.83 | 354.61±33.34 | 236.61±43.49 | 386.77±30.09 | 464.07±6.61 |
| | Avg. CV-G ↓ | 0.33±0.37 | 9.96±4.54 | 12.72±9.91 | 11.18±7.46 | 60.39±4.32 |
| Ant-Run | Avg. R ↑ | 660.88±4.82 | 615.73±91.99 | 594.75±172.35 | 636.06±6.78 | 629.83±7.84 |
| | Avg. CV ↓ | 3.13±1.67 | 8.47±3.55 | 23.69±30.42 | 5.16±1.59 | 0.22±0.17 |
| | Avg. R-G ↑ | 660.07±5.26 | 626.27±84.61 | 592.50±173.01 | 620.46±9.99 | 605.07±10.63 |
| | Avg. CV-G ↓ | 3.25±1.48 | 7.76±11.83 | 22.90±9.39 | 6.73±2.32 | 0.03±0.06 |

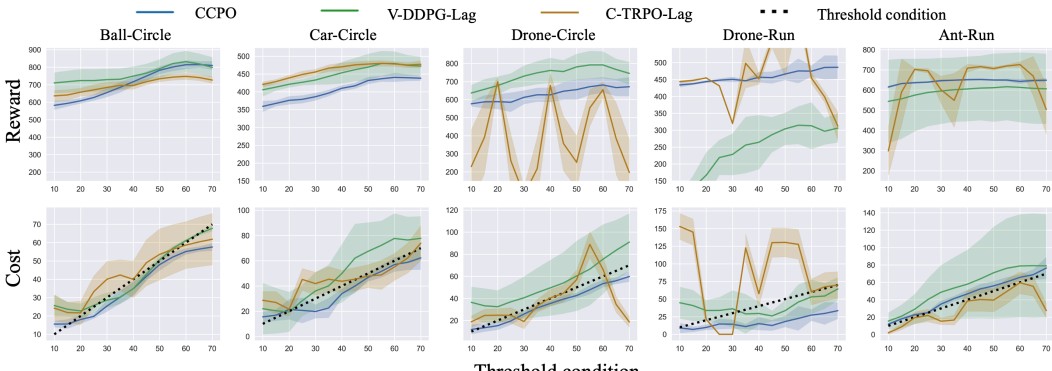

Figure 2: Results of zero-shot adaption to different cost returns. Each column is a task. The x-axis is the threshold condition. The first row shows the evaluated reward, and the second row shows the evaluated cost under different target costs. All plots are averaged among 5 random seeds and 50 trajectories for each seed. The solid line is the mean value, and the light shade represents the area within one standard deviation. We train the versatile agent with behavior policy conditions $\tilde{\mathcal{E}} = \{20, 40, 60\}$, and evaluate it on $\mathcal{E} = \{10, 15, ..., 70\}$.

and `Car-Circle`, their cost violation values are significantly larger than the proposed `CCPO` cost. In the `Drone-Circle`, `Drone-Run`, and `Ant-Run` tasks characterized by highly-nonlinear robot dynamics, all the baseline methods exhibit poor generalization when being exposed to different thresholds, thus leading to high cost violation values. This limitation arises due to the inadequacy of utilizing only a limited number of behavior policies for versatile policy training in tasks with high-dimensional observation and action spaces. Insufficient conditions prevent the actor from effectively distinguishing between different constraint conditions, thus resulting in large performance variance even for behavior policies, and large cost violations on unseen thresholds. **The poor safety**

**performance when generalizing to different threshold conditions indicates the necessity of studying versatile safe RL methods.**

Second, although the linear combination of single-threshold policies performs well on `Ball-Circle` and `Car-Circle` with simple robot dynamics, it can hardly handle `Drone-Circle`, `Drone-Run`, and `Ant-Run` tasks – the interpolation method has a significant reward drop at unseen thresholds. It is because, for `Ball-Circle` and `Car-Circle` task, the action dimensionality is low and the dynamics are simple, thus directly applying linear combination may work in these settings. However, for `Drone-Circle` and `Drone-Run`, the high nonlinearity in agent dynamics and the large action space will make this naive approach fail to perform well. **These results show that the concepts from the control theory that the safety-critical control component is proportional to the conservativeness level can not be directly used in versatile safe RL with high-dimensional settings,** which further indicates the necessity of the versatile safe RL method.

Finally, from Table 1, we can clearly see that **our proposed CCPO method learns a versatile safe RL policy that can generalize well to unseen thresholds** with low cost-violations and high rewards. Also, from Figure. 2, we can observe the performance of the CCPO method has a smooth relation with respect to threshold conditions, which indicates it can efficiently encode the threshold conditions into the versatile safe RL agent.

## 5.2 Evaluation of $\epsilon$-sampling efficiency

The proposed algorithm is $\epsilon$-sampling-efficient as it satisfies the first requirement **thresholds sampling efficiency** mentioned in section 3.2: it is able to train the versatile safe RL agent with limited behavior policies for data collection. In this experiment, we aim to answer the question: How $\epsilon$-efficient CCPO is compared to exhaustively training the safe RL agents? We compare our method with C-TRPO, which is the strongest baseline method as shown in Table. 1 with different behavior policy set $\tilde{\mathcal{E}} = \{20, 40, 60\}$ and $\tilde{\mathcal{E}}' = \{20, 30, 40, 50, 60, 70\}$, where $|\tilde{\mathcal{E}}'| = 2|\tilde{\mathcal{E}}|$. The algorithms are evaluated on threshold conditions $\mathcal{E} = \{10, 15, ..., 70\}$, and the averaged performance is reported in Table. 2.

We observe that CCPO exhibits significant $\epsilon$-efficiency over C-TRPO-Lag: CCPO outperforms C-TRPO-Lag significantly in terms of both safety and reward, even with a smaller set of behavior policy conditions. This comparison showcases the remarkable $\epsilon$-efficiency of our approach. In fact, our method demonstrates at least 2 times greater $\epsilon$-sampling efficiency compared to exhaustively training safe RL agents using C-TRPO-Lag.

Table 2: $\epsilon$-sampling efficiency evaluation. $\uparrow$: the higher reward, the better. $\downarrow$: the lower constraint violation (minimal 0), the better. The models are evaluated on a series of threshold conditions and we report the averaged reward and constraint violation values on all evaluation thresholds and generalized thresholds. Each value is reported as mean ± standard deviation for 50 episodes and 5 seeds. We shade the safest agent with the lowest averaged cost violation value.

| Algorithm | Stats | Ball-Circle | Car-Circle | Drone-Circle | Drone-Run | Averaged Score |
|---|---|---|---|---|---|---|
| CCPO with $\tilde{\mathcal{E}}$ | Avg. R $\uparrow$ | 710.86±20.47 | 406.06±6.30 | 630.55±40.03 | 458.69±12.98 | 551.54 |
| | Avg. CV $\downarrow$ | 0.59±0.31 | 1.60±0.91 | 0.32±0.38 | 0.23±0.25 | 0.69 |
| C-TRPO with $\tilde{\mathcal{E}}$ | Avg. R $\uparrow$ | 699.38±1.94 | 461.14±1.39 | 380.77±18.62 | 461.70±4.91 | 500.75 |
| | Avg. CV $\downarrow$ | 4.50±0.08 | 7.84±1.71 | 6.55±1.95 | 47.97±3.49 | 16.72 |
| C-TRPO with $\tilde{\mathcal{E}}'$ | Avg. R $\uparrow$ | 682.94±8.08 | 458.13±2.22 | 411.91±8.95 | 472.89±2.65 | 506.47 |
| | Avg. CV $\downarrow$ | 2.66±0.37 | 11.90±2.12 | 5.20±0.81 | 30.20±2.47 | 12.49 |

## 5.3 Ablation Study

To study the influence of VVE, and CVI components of CCPO introduced in the section 4.1 and 4.2, we conduct an ablation study by removing each component from the full CCPO algorithm. The ablation experiment results are shown in Table. 3. We shade the safest agent with the lowest averaged cost violation value. We can also observe significant safety performance (cost violation) degradation and task performance (reward) drop if we remove the VVE module (versatile critic learning) or CVI module (versatile actor learning) since they help us learn versatile Q functions more accurately and

improve the generalizability of learned actors. Removing them will result in the bad estimation of for state-action pair value function and lead to poor policy generalization capability.

Table 3: Ablation study of removing the versatile value function estimation (VVE), and the conditioned variational inference (CVI). ↑: the higher reward, the better. ↓: the lower constraint violation (minimal 0), the better. Each value is reported as mean ± standard deviation for 50 episodes and 5 seeds. Each value is reported as mean ± standard deviation.

| Algorithm | Stats | Ball-Circle | Car-Circle | Drone-Circle | Drone-Run | Ant-Run |
|---|---|---|---|---|---|---|
| CCPO (Full) | Avg. R ↑ | 710.86±20.47 | 406.06±6.30 | 630.55±40.03 | 458.69±12.98 | 660.88±4.82 |
| | Avg. CV ↓ | 0.59±0.31 | 1.60±0.91 | 0.32±0.38 | 0.23±0.25 | 3.13±1.67 |
| | Avg. R-G ↑ | 699.04±20.48 | 401.53±5.59 | 625.51±40.12 | 455.64±11.83 | 660.07±5.26 |
| | Avg. CV-G ↓ | 0.83±0.42 | 1.49±0.38 | 0.47±0.55 | 0.33±0.37 | 3.25±1.48 |
| CCPO w/o VVE | Avg. R ↑ | 674.55±17.81 | 370.42±14.38 | 426.47±49.30 | 417.84±8.24 | 428.59±88.39 |
| | Avg. CV ↓ | 0.60±0.41 | 6.42±0.85 | 8.67±1.45 | 3.28±2.86 | 10.66±11.81 |
| | Avg. R-G ↑ | 670.61±14.18 | 364.5±14.51 | 416.83±47.46 | 413.28±9.04 | 434.59±83.89 |
| | Avg. CV-G ↓ | 0.73±0.36 | 5.64±0.92 | 7.74±1.36 | 3.33±3.16 | 12.01±10.08 |
| CCPO w/o CVI | Avg. R ↑ | 641.33±40.14 | 387.31±5.76 | 520.70±42.18 | 386.81±39.44 | 465.80±31.78 |
| | Avg. CV ↓ | 1.44±0.72 | 1.66±0.79 | 2.36±2.67 | 0.81±0.76 | 3.51±0.93 |
| | Avg. R-G ↑ | 623.17±41.42 | 383.24±6.30 | 519.05±36.31 | 388.69±35.35 | 465.36±32.20 |
| | Avg. CV-G ↓ | 1.78±0.70 | 2.17±1.09 | 2.73±3.03 | 1.15±1.08 | 3.96±1.01 |

## 6 Conclusion

In this study, we pioneered the concept of versatile safe reinforcement learning (RL), presenting the Constraint-Conditioned Policy Optimization (CCPO) algorithm. This approach adapts efficiently to different and unseen cost thresholds, offering a promising solution to safe RL beyond pre-defined constraint thresholds. With its core components, Versatile Value Estimation (VVE) and Conditioned Variational Inference (CVI), CCPO facilitates zero-shot generalization for constraint thresholds. Our theoretical analysis further offers insights into the constraint violation bounds for unseen thresholds and the sampling efficiency of the employed behavior policies. The extensive experimental results reconfirm that CCPO effectively adapts to unseen threshold conditions and is much safer and more data-efficient than baseline methods. The limitations include that our method would be more computationally expensive than the primal-dual-based safe RL approaches due to the convex optimization problems in the constraint-conditioned E-step. One potential negative social impact is that the misuse of this work in safety-critical scenarios can cause unexpected damage. We hope our findings can inspire more research in studying the generalization capability in safe RL.

## 7 Acknowledgment

The work is partially supported by Google Deepmind with an unrestricted grant. The authors also want to acknowledge the support from the National Science Foundation under grants CNS-2047454.

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
