# OpenReview forum: "Constraint-Conditioned Policy Optimization for Versatile Safe Reinforcement Learning"
_NeurIPS.cc/2023/Conference — NeurIPS 2023 poster_

### Official Review · Reviewer_tMdC · 2023-07-01

**Soundness:** 4 excellent
**Presentation:** 4 excellent
**Contribution:** 3 good
**Rating:** 7
**Confidence:** 4

**Summary:**

1. This paper introduces the Conditioned Constrained Policy Optimization (CCPO) framework, consisting of two modules: (1) Versatile Value Estimation (VVE) for approximating value functions under unseen threshold conditions, and (2) Conditioned Policy Inference (CPI) for encoding arbitrary constraint thresholds during policy optimization.
2. The proposed method can enable learning versatile safe policies that can adapt to varying safety constraint requirements during deployment without retraining.

**Strengths:**

1. training efficiency and zero-shot adaptation capability: the trained policy can adapt to varying safety constraint requirements during deployment without retraining.
2. The theoretical analysis for the q-function estimation and constraint violations.
3. The extensive experiments are implemented to demonstrate the effectiveness of the method.

**Weaknesses:**

See questions.

**Questions:**

1. How reasonable/strong is assumption 2? Especially if latent vectors are highly dimensional. In experiments, what model do you use to estimate the latent vector z(\epsilon)?
2. What is the main technical challenge in extending CVPO [30] to the constraint-conditional scenarios?
3. How does this method scale when there are multiple constraints?
4. How does this work compare with the following missing literature with similar topics:
Meta CMDP:
Khattar, V., Ding, Y., Sel, B., Lavaei, J., & Jin, M. (2022, September). A CMDP-within-online framework for Meta-Safe Reinforcement Learning. In The Eleventh International Conference on Learning Representations.
Non-stationary CMDP:
Ding, Y., & Lavaei, J. (2023, June). Provably efficient primal-dual reinforcement learning for CMDPs with non-stationary objectives and constraints. In Proceedings of the AAAI Conference on Artificial Intelligence.
Wei, H., Ghosh, A., Shroff, N., Ying, L., & Zhou, X. (2023, April). Provably Efficient Model-Free Algorithms for Non-stationary CMDPs. In International Conference on Artificial Intelligence and Statistics (pp. 6527-6570). PMLR.

---

> ### Author Rebuttal · Authors · 2023-08-08
>
> We thank the reviewer for your time, valuable feedback, and acknowledgment of our contribution. We address the concerns as follows.
>
> > Q1: How reasonable/strong is assumption 2? Especially if latent vectors are highly dimensional. In experiments, what model do you use to estimate the latent vector $z(\epsilon)$?
>
> Thank you for your question. Assumption $2$ is considered mild. We provide some verification and visualization results of $Q(\cdot | \epsilon)$ estimation in Appendix D.1, which support the reasonableness of using this assumption for training. In practice, we employ a small Neural-Network model, as shown in the illustrations in Appendix C.1.
>
> > Q2: What is the main technical challenge in extending CVPO [30] to the constraint-conditional scenarios?
>
> Thank you for your question. The main technical challenge for CVPO is the estimation of the Q function $Q(\cdot|\epsilon)$ for arbitrary threshold $\epsilon$, which has been addressed in the **VVE** module of this work.
>
> > Q3: How does this method scale when there are multiple constraints?
>
> Thank you for your insightful question. We will leave the multi-constraint versatile safe RL in the future work as challenges arise from two factors: (1) Higher dimension in $z_f(\epsilon)$: Since $\epsilon$ is now a vector, learning $z_f$ becomes more challenging. (2) Multi-constraint safe RL itself is more complex than the single-constraint scenario.
>
>
> > Q4: How does this work compare with the following missing literature with similar topics...
>
> Thank you for your feedback. The comparison is as follows.
>
> **The Meta CMDP paper:** Khattar, V., Ding, Y., Sel, B., Lavaei, J., & Jin, M. (2022, September). A CMDP-within-online framework for Meta-Safe Reinforcement Learning. In The Eleventh International Conference on Learning Representations. This work focuses on Multi-task Safe RL, involving tasks with different initialization conditions, and dynamic environments, rather than varying constraint thresholds.
>
> **The Non-stationary CMDP papers:** Ding, Y., & Lavaei, J. (2023, June). Provably efficient primal-dual reinforcement learning for CMDPs with non-stationary objectives and constraints. In Proceedings of the AAAI Conference on Artificial Intelligence; and Wei, H., Ghosh, A., Shroff, N., Ying, L., & Zhou, X. (2023, April). Provably Efficient Model-Free Algorithms for Non-stationary CMDPs. In International Conference on Artificial Intelligence and Statistics (pp. 6527-6570). PMLR. These works address safe RL problems in non-stationary environments, where cost/reward functions change during an episode. In contrast, our work focuses on settings with fixed reward/cost functions and varying cost threshold conditions. Exploring versatile safe agent learning in non-stationary environments could be an interesting direction for future research.
>
>
> We will add the additional discussion in the revision.

---

### Official Review · Reviewer_TpqY · 2023-07-03

**Soundness:** 3 good
**Presentation:** 2 fair
**Contribution:** 2 fair
**Rating:** 7
**Confidence:** 3

**Summary:**

The paper addresses Safe Reinforcement Learning with varying safety constraint requirements (Versatile Safe Reinforcement Learning). The paper proposes to learn a versatile Q function, which is decomposed as the product of feature functions of state and action,  and the latent features of the constraint thresholds. The Q function will be used in a variational EM algorithm to find the constraint-satisfying policy that maximizes the return. The proposed method shows good safety and decent returns in experimental robotic tasks such as Run and Circle, outperforming simple constraint-conditioned and linear combination baselines.

**Strengths:**

- The target problem is interesting and seems novel in the setting of Safe RL
- The paper has relevant theoretical analysis to back up the method

**Weaknesses:**

- The writing is unclear, especially in the pretraining phase. It is better to have an algorithm in the main text to summarize the process and connect the components in the method. It is hard to connect Fig. 1 to the method section. For example, where are the two bottom boxes in the pretraining phase described in the method section?
- The experiment result overall is not really good. Although the safety criteria show good performance, the reward counterpart is not. In many tasks, the proposed method underperforms the baselines regarding the rewards. How to ensure that the trade-off is acceptable and the behaviour is desirable in Safe RL setting? Overall, more tasks should be tested to confirm if the lower performance is a consistent issue in the approach. Furthermore, there are claims on sample efficiency, yet no report on it can be found in the main text. It should be noted that running time should be considered as well because as admitted in the discussion, the method can be slow due to complicated optimization steps.


**Questions:**

See Weakness.

**Limitations:**

The method is complicated with different stages of optimization. Can it work in higher dimensional domains such as visual state space?

---

> ### Author Rebuttal · Authors · 2023-08-08
>
> We thank the reviewer for your time and valuable feedback. We address the concerns as follows.
>
> > W1: The writing is unclear, especially in the pretraining phase...
>
> Thank you for your valuable suggestions. While we placed the detailed algorithm diagram in the appendix due to page constraints, we wholeheartedly concur that having an algorithm diagram in the main text for summarizing the process and interconnecting method components is advantageous. To cater to this, we offer a concise algorithm diagram in the attached PDF. In this diagram, lines 1-2 correspond to the VVE module, while lines 3-15 correspond to the CVI procedure. Specifically, during the pre-training phase, the selection of threshold conditions for training is denoted as the behavior policy condition set (line 5). In the fine-tuning stage, the choice shifts to a larger condition set (line 5). In Figure 1, "Versatile Safe RL Agent" signifies the policy incorporating thresholds as conditions, whereas "Constraint-Conditioned Behavior Policy Learning" encompasses the CVI process with behavior policy conditions (lines 3-15 in the new algorithm diagram). We will add the algorithm diagram in the revision.
>
>
> > W2.1: ... Although the safety criteria show good performance, the reward counterpart is not...
>
> Thank you for your valuable feedback. We understand the reviewer's confusion that the rewards of baselines in some tasks are slightly higher than our method's. However, we would like to gently point out that this is a common trade-off between task utility and safety performance in the safe RL literature -- the **optimal** policy under safety constraints might not be the most rewarding one [R-1, R-2, R-3].
> To be more concrete, the evaluation criteria in safe RL are:
> 1) method A is better than method B if A achieves better safety performance than B.
> 2) If both A and B satisfy constraints, the one with the higher reward is better.
>
> Therefore, we can conclude that CCPO outperforms baselines in most experiments. We will make the evaluation criteria clear in the revision to avoid confusion.
>
> > W2.2: Claims on sample efficiency...
>
> Thank you for your valuable feedback. Due to the page limit, we initially placed the discussion about $\epsilon$-sampling efficiency in Appendix D.4. However, we acknowledge the inconvenience this may cause and will address this by incorporating the discussion into the main context in the revision. Below are the evaluation results.
>
> The proposed algorithm effectively demonstrates $\epsilon$-sampling efficiency, as it meets the first requirement outlined in **Section 3.2: Thresholds Sampling Efficiency**: enabling training of a versatile safe RL agent with a limited set of behavior policies for data collection. We compare our approach with C-TRPO, the strongest baseline method, as highlighted in Table 1. We evaluate algorithms using different behavior policy sets, specifically $\tilde{\mathcal{E}} = \{20, 40, 60\}$ and $\tilde{\mathcal{E}}^{\prime} = \{20, 30, 40, 50, 60, 70\}$, where $|\tilde{\mathcal{E}}^{\prime}| = 2 \times |\tilde{\mathcal{E}}|$. The evaluation encompasses threshold conditions $\mathcal{E} = \{10, 15, ..., 70\}$, and we report the averaged performance outcomes in the subsequent table.
>
>
> | Tasks | stats | BC | CC |  DC | DR | Average |
> | :------: | :------: | :------: | :------: | :------: | :------: | :------: |
> | CCPO with $\tilde{\mathcal{E}}$ | Avg. R $\uparrow$ | 710.86 | 406.06 | 630.55 | 458.69 | **551.54** |
> | | Avg. CV $\downarrow$ | 0.59 | 1.60 | 0.32 | 0.23 | **0.69** |
> | C-TRPO with $\tilde{\mathcal{E}}$ | Avg. R $\uparrow$ | 699.38 | 461.14 | 380.77 | 461.70 | 500.75 |
> | | Avg. CV $\downarrow$ | 4.50 | 7.84 | 6.55 | 47.97 | 16.72 |
> | C-TRPO with $\tilde{\mathcal{E}}^{\prime}$ | Avg. R $\uparrow$ | 682.94 | 458.13 | 411.91 | 472.89 | 506.47 |
> | | Avg. CV $\downarrow$ | 2.66 | 11.90 | 5.20 | 30.20 | 12.49 |
>
> We observe that **CCPO Exhibits Significant $\epsilon$-Efficiency Over C-TRPO:** CCPO outperforms C-TRPO significantly in terms of both safety and reward, even with a smaller set of behavior policy conditions. This comparison showcases the remarkable $\epsilon$-efficiency of our approach. In fact, our method demonstrates at least 2 times greater $\epsilon$-sampling efficiency compared to exhaustively training safe RL agents using C-TRPO.
>
> > W2.3: It should be noted that running time should be considered as well because as admitted in the discussion, the method can be slow due to complicated optimization steps.
>
> Thank you for your valuable feedback and detailed discussion. The running time is presented below.
>
> **Running time:** All the experiments are run on the AMD EPYC 7713 64-Core Processor without GPU acceleration. We report the averaged running time based on the experiment logs in the following table. All the experiments can be done within 24h, and the training can be further accelerated if GPUs are available.
>
> | Tasks | BC | CC | DC | DR | AR |
> | :------: | :------: | :------: | :------: | :------: | :------: |
> | CCPO | 10h | 10h | 22h | 15h | 23h |
> | V-DDPG | 6h | 7h | 17h | 14h | 13h |
>
> > L1: The method is complicated...
>
> Thank you for your feedback. The proposed CCPO method has the potential to extend to visual-input tasks. However, state feature extracting may be necessary since it can help in the versatile Q function learning and the policy optimization process. We leave it as a future work.
>
> **Reference**
>
> [R-1] Yu, Haonan, Wei Xu, and Haichao Zhang. "Towards safe reinforcement learning with a safety editor policy." Advances in Neural Information Processing Systems 35 (2022): 2608-2621.
>
> [R-2] Stooke, Adam, Joshua Achiam, and Pieter Abbeel. "Responsive safety in reinforcement learning by pid lagrangian methods." International Conference on Machine Learning. PMLR, 2020.
>
> [R-3] Liu, Zuxin, et al. "On the Robustness of Safe Reinforcement Learning under Observational Perturbations." The Eleventh International Conference on Learning Representations. 2022.

---

> > ### Comment · Reviewer_TpqY · 2023-08-16
> > **Rebuttal response**
> >
> > Thank you for answering my questions. The rebuttal mostly cleared my concerns. Hence, I will raise my score to 7. Regarding the point "we can conclude that CCPO outperforms baselines in most experiments", I think the authors need to report the results by following the "evaluation criteria in safe RL". For example, in the result table, methods that satisfy constraints should be grouped and compared in terms of rewards.  It is best if more tasks are considered to ensure that the trade-off is acceptable in various settings.

---

> > > ### Author Response · Authors · 2023-08-17
> > > **Thank you!**
> > >
> > > We sincerely appreciate the reviewer's recognition of our paper and raising the score. Thanks again for the constructive comments!

---

### Official Review · Reviewer_5QUs · 2023-07-06

**Soundness:** 3 good
**Presentation:** 2 fair
**Contribution:** 3 good
**Rating:** 7
**Confidence:** 2

**Summary:**

Safe reinforcement learning approaches have so far focused on the case where
cost thresholds are static and known a priori. In contrast, this work addresses
RL environments where cost constraints may change over time. To do this, the
proposed algorithm learns a policy which is conditioned on the given cost
threshold, allowing the policy to adapt to new cost thresholds at test time.
The policy is derived from a pair of learned Q functions, one for rewards and
one for costs. The Q functions are decomposed into a feature for state-action
pairs and a feature for the safety threshold. The safety threshold feature uses
a relatively low-dimensional model in order to provide better generalization
without needing to see a large set of thresholds at training time, making
training more efficient. In experiments, the proposed algorithm outperforms
Lagrangian approaches to safety in a suite of relatively high-dimensional
benchmarks.

**Strengths:**

The problem of versatlie safe RL is interesting, potentially important, and (to
my knowledge) novel. The threshold in CMDP-based safe RL work is often
arbitrary, so it seems useful to be able to modify that threshold without
retraining the system. Moreover, there may be some environments where the cost
threshold changes over time.

The proposed high-level approach is a very elegant solution to this problem. I
particularly like the idea of separating out the effect of $\epsilon$ into a
lower-parameter model which should generalize better than a full neural model.
The way this allows for efficient training without sacrificing generalizability
to new safety thresholds is quite impressive. Similarly, separating out Q-value
learning from policy construction is an interesting way to generalize while
maintaining the power of neural learning.

The experiments are fairly extensive and show impressive results. In
particular, CPPO demonstrates much better safety than other approaches on most
benchmarks, while maintaining comparable performance. I think Figure 2 is a
good illustration of the advantages of CPPO, showing how the number of observed
cost closely tracks the allowed cost. This suggests that CPPO is using all of
its available safety budget to achieve high rewards.

**Weaknesses:**

I think the problem statement could do with a bit more motivation. While I do
believe the versatile safe RL problem is interesting, it's not clear to me how
often these kinds of problems actually arise. The given example of a car
operating in highway vs. urban environments provides some intuition, but I'm
not clear on what the cost signal is that would be changing in that case. An
extra example or more explanation might help clarify this.

I found the development in Section 4 to be difficult to follow. I'm not very
familiar with the work on RL as inference, so maybe this explanation is
sufficient for more expert readers. For less expert readers, I think it would
help to add a bit more intuition or maybe some examples if feasible.

For the experiments, I think it would help to include some baselines from the
safe RL literature beyond basic Lagrangian approaches. For example, you might
compare with CPO or Conservative Safety Critics (among others). These
approaches are generally tailored to a single threshold, but a comparison could
be made by training them with a strict threshold then using a more relaxed
threshold at test time. In principle, CPPO should achieve a higher reward since
it is able to take advantage of the increased safety budget.

**Questions:**

Do you have any intiution about why TRPO-Lag achieves better safety on Ant-Run
than CCPO? Is there some feature of the benchmark or the algorithm which makes
this a particularly challening case for CCPO?

**Limitations:**

The authors have adequately addressed some limitations and potential societal
impacts in the paper.

---

> ### Author Rebuttal · Authors · 2023-08-08
>
> We gratefully thank the reviewer for your time, valuable feedback, and acknowledgment of our contribution. We address your concerns as follows.
>
> > W1: The given example of a car operating in highway vs. urban environments provides some intuition, but I'm not clear on what the cost signal is that would be changing in that case. An extra example or more explanation might help clarify this.
>
> We sincerely appreciate your interest in our proposed versatile safe RL settings. Here is an extra example: for a legged delivery robot, we can define the reward as the walking speed and the cost as the shaking amplitude. Intuitively, stricter constraints on cost typically lead to more conservative behavior (low-speed motion) and lower task rewards. In this case, the robot can adapt to a larger cost threshold for carrying non-fragile than carrying fragile cargo to maximize working efficiency.
>
> > W2: ... I think it would help to add a bit more intuition or maybe some examples in Section 4 if feasible.
>
> Thank you for your great suggestion. We fully agree that it would be desirable to provide some additional explanations. Here, we present a toy-like example for **Section 4.2. Conditional Variational Inference**. Consider the task illustrated in Figure R-2, where the action space is discrete and has a dimension of 2 (representing the acceleration and angular velocity of the unicycle). The key idea of this section is to first calculate the optimal distribution $q^*(a | s, \epsilon_i)$ for a state $s$ and sampled actions $a$ in the **Constraint-Conditioned E-step**, and subsequently optimize the policy $\pi(\cdot|\epsilon_i)$ based on $q^*(a | s, \epsilon_i)$ using supervised learning in the **Versatile M-step**.
>
>
> As shown in the Figure. R-3, for a given state $s$, we sample all possible actions and calculate the corresponding optimal distribution $q^*$ within the feasible distribution $q$ set. The optimization variable $q$ represents the probabilities of taking each sampled action. In the **Constraint-Conditioned E-step**, when dealing with different thresholds $\epsilon_1 = 10$ and $\epsilon_2 = 20$, the feasible $q$ set calculated based on $Q_c(\cdot| \epsilon_i)$ may differ, as illustrated in Figure R-3 $(a)$ and $(c)$. Consequently, the optimal distribution $q(\cdot|\epsilon_i)$ will also vary - in the looser condition $\epsilon_2=20$, the agent can take more aggressive actions to achieve higher rewards, as illustrated in Figure R-3 $(b)$ and $(d)$. After obtaining the optimal distribution $q^*(\cdot|\epsilon_i)$, in the **Versatile M-step**, we can improve the Evidence Lower Bound (ELBO) of $\pi(\cdot|\epsilon_i)$ with respect to $q^*(\cdot|\epsilon_i)$ for policy optimization, as shown in Eq. (12) in the main text.
>
>
>
> We welcome any additional comments you may have on this example. Thank you for your feedback!
>
> > W3: More baselines...
>
> Thank you for your valuable feedback. We have made modifications to CPO (via policy linear combination) creating the Versatile CPO, and conducted experiments in all five tasks. The following table presents the comparison results between Versatile CPO and our proposed CCPO. It is evident that the proposed CCPO method outperforms Versatile CPO in terms of both constraint satisfaction and reward efficiency.
>
> | Tasks | stats | BC | CC | DC | DR | AR | Average |
> | :------: | :------: | :------: | :------: | :------: | :------: | :------: | :------: |
> |  Versatile CPO | Avg. R $\uparrow$ | 686.08 ± 2.58 | 455.87 ± 1.51 | 565.98 ± 5.71 | 304.09 ± 32.17 | 602.05 ± 1.66 | 522.81
> |  | Avg. CV $\downarrow$ | 1.77 ± 0.67 | 4.50 ± 2.08 | 6.32 ± 2.32 | 11.82 ± 5.85 | 2.19 ± 0.63 | 6.60
> |  | Avg. R-G $\uparrow$ | 676.27 ± 2.32 | 454.62 ± 1.97 | 559.71 ± 7.32 | 300.85 ± 32.41 | 592.89 ± 2.68 | 516.87
> |  | Avg. CV-G $\downarrow$ | 2.09 ± 0.59 | 4.94 ± 2.37 | 6.75 ± 2.47 | 13.07 ± 5.44 | 2.28 ± 0.73 | 5.83
> | CCPO | Avg. R $\uparrow$ | 710.86 ± 20.47 | 406.06 ± 6.30 | 630.55 ± 40.03 | 458.69 ± 12.98 | 660.88 ± 4.82 | **573.41**
> |  | Avg. CV $\downarrow$ | 0.59 ± 0.31 | 1.60 ± 0.91 | 0.32 ± 0.38 | 0.23 ± 0.25 | 3.13 ± 1.67 | **1.17**
> |  | Avg. R-G $\uparrow$ | 699.04 ± 20.48 | 401.53 ± 5.59 | 625.51 ± 40.12 | 455.64 ± 11.83 | 660.07 ± 5.26 | **568.36**
> |  | Avg. CV-G $\downarrow$ | 0.83 ± 0.42 | 1.49 ± 0.38 | 0.47 ± 0.55 | 0.33 ± 0.37 | 3.25 ± 1.48 | **1.27**
>
> > Q1: Intuition about why TRPO-Lag achieves better safety on Ant-Run than CCPO.
>
> Thank you for your valuable feedback. Let's briefly introduce the Ant-Run task and analyze the results. In Ant-Run, the reward is defined by the distance moved along one direction, and the cost signal is sparse (i.e., 0-1 cost), where the agent incurs a cost once it exceeds the speed limit or goes beyond the boundary. The dynamics of the Ant are complicated, with the observation space dimension being $33$, containing the ego position, joint angles, etc. The action space dimension is $8$, encompassing the torque of each joint.
>
>
> Considering the TRPO-Lag baseline, we observe in Figure 2 of the main text that there are significant reward performance drops at some unseen thresholds (e.g., 10, 15, 25, 30, 35, 65, ...). These drops indicate that under these conditions, the walking ability of Ant is poor. Consequently, the risk of the Ant going beyond the speed limit is also low, which explains why TRPO-Lag achieves higher safety performance in Table 1. On the other hand, our proposed CCPO algorithm exhibits stable and smooth reward performance with respect to different threshold conditions. Moreover, the cost returns in CCPO consistently adhere to the target thresholds on nearly every unseen threshold with low coefficient of variation (CV) and variance, showcasing CCPO's more reliable and stable performance.

---

> > ### Comment · Reviewer_5QUs · 2023-08-10
> >
> > Thank you for the response. The extra motivating example in W1 is helpful, and the extra results in W3 are convincing. I will keep my score in favor of acceptance.

---

> > > ### Author Response · Authors · 2023-08-17
> > > **Thank you!**
> > >
> > > We sincerely appreciate the reviewer's recognition of our paper. Thanks again for the constructive comments!

---

### Official Review · Reviewer_hfso · 2023-07-07

**Soundness:** 3 good
**Presentation:** 3 good
**Contribution:** 3 good
**Rating:** 7
**Confidence:** 4

**Summary:**

The authors introduce the versatile safe RL problem, in which we want to maximize the expected reward, subject to an expected constraint, but for a range of constraint thresholds $\epsilon\in \mathcal E$ (compared to the traditional safe RL problem, which only aims to find a solution for one $\epsilon$). This is an interesting problem setting, with good motivation and potential real world application in learning a policy that works under many circumstances. The authors also provide an algorithm CVPO,  that nicely combines the control as inference framework with the EM framework to improve the likelihood of optimality while trying to stay within a feasible set.

**Strengths:**



**Weaknesses:**



**Questions:**


On Page 4, Line 134 - what is $K_f$? is it a constant for $f \in \{r, c\}$? Similarly, on line 146, I understand that $Poly(\epsilon, p)$ is a $p$-degree polynomial of $\epsilon$, but shouldn't it have a vector output since $z_f^*$ is a vector? Also, on line 158, why is it okay to normalize epsilon to [0, 1] since realistically constraint thresholds can be arbitrary?

On Page 5, Line 197, the authors say - "The key strength of using the variational inference framework lies in its ability to encode arbitrary threshold conditions during policy learning, as shown in (7), a feat that is challenging for other methods, such as those based on primal-dual algorithms." Could you elaborate why this is challenging for other methods?

For testing generalization to unseen thresholds, the authors train on data from $\epsilon$=20,40,60 and test with $\epsilon$=10,20,30,40,50,60,70. This is relatively close to the thresholds in the training data. It would be nice to see this algorithm trained with thresholds in [20, 30, 40] and tested with thresholds in [50, 60, 70]. Ideally, the algorithm may be able to provide feasible policies, but it is also important to test if the reward achieved is highest possible. If it is able to handle thresholds well outside the training set, that would be a good thing and it will show that the algorithm is quite versatile. If it is unable to do so, then, it should be discussed a bit further i.e. what unseen thresholds can we realistically expect good performance on?

Suggestions related to language and typos:
1. On Page 5 - "We utilize the safe RL as inference framework ..." - since this is proposed by this work, it would be better to say "we introduce the ...".

EDIT (16 Aug 2023): Updated score from 6->7.

**Limitations:**

---

> ### Author Rebuttal · Authors · 2023-08-08
>
> We thank the reviewer for your time and valuable feedback. We address the concerns as follows.
>
> > Q1: On Page 4, Line 134 - what is $K_f$ is it a constant for $f \in r, c$ Similarly, on line 146, I understand that $Poly(\epsilon, p)$ is a $p$-degree polynomial of $\epsilon$, but shouldn't it have a vector output since $z_f$ is a vector? Also, on line 158, why is it okay to normalize $\epsilon$ to [0, 1] since realistically constraint thresholds can be arbitrary?
>
> Thank you for your questions. Yes, $K_f$ is a constant for $f \in r, c$, and each element of $z_f$ is a $p$-degree polynomial of $\epsilon$. In this work, we consider finite-step trajectories and bounded costs for each step. Consequently, we have a maximum cost value for each trajectory. To simplify the process, we can normalize $\epsilon$ by dividing it by the upper bound for trajectory cost. All the experiment tasks, as well as many common safe RL benchmarks such as safety-gym [R-1], comply with these assumptions (finite-step, bounded cost value). We apologize if the formulation has caused any confusion, and we will further refine it in the revision.
>
> > Q2: On Page 5, Line 197, the authors say - "The key strength of using the variational inference framework lies in its ability to encode arbitrary threshold conditions during policy learning, as shown in (7), a feat that is challenging for other methods, such as those based on primal-dual algorithms." Could you elaborate why this is challenging for other methods?
>
> Thank you for your insightful question. Taking Lagrangian-based safe RL algorithms as an example, the key component of such algorithms is to find a proper $\lambda$ balancing the trade-off between reward and cost satisfaction, i.e., $\lambda$ for $V_r - \lambda V_c$ in the policy loss. The multiplier $\lambda$ is a function of the policy $\pi$. One of the most common approaches is using the PID method [R-2] to update $\lambda$ with the cost difference $(\Sigma c_t-\epsilon)$. Such methods present a major challenge for versatile safe RL: **Lagrangian multiplier update for unseen thresholds.** Let $\epsilon_1$ represent one behavior policy condition, and $\epsilon_2 < \epsilon_1$ represents another unseen threshold for adaptation. Let $\lambda_1$ and $\lambda_2$ be the corresponding Lagrangian multipliers. Suppose we update $\lambda_2$ using trajectory data from the behavior policy $\pi(\cdot | \epsilon_1)$; since $\Sigma c_t$ converges to $\epsilon_1$, then $\Sigma c_t - \epsilon_2$ will never converge to $0$, which makes $\lambda_2$ overestimated. This could make the policy $\pi(\cdot|\epsilon_2)$ overly conservative.
>
> With the CCPO method, we do not encounter such problems since we encode the threshold condition in the constraint-conditioned E-step by setting constraints in an off-policy fashion while calculating the optimal variational distribution $q^*(\cdot | \epsilon)$. This constraint is flexible for arbitrary $\epsilon$ as long as we have obtained a good estimation of the Q-function $Q_c(\cdot | \epsilon)$, which is accomplished in the **VVE** module of this work.
>
> > Q3: For testing generalization to unseen thresholds...
>
> Thank you for your valuable feedback. The experiment results for CCPO with behavior policy condition $[20, 30, 40]$ and evaluation on $[10, 15, ..., 70]$ are shown in Figure R-1 and the following table. Each value is reported as mean ± standard deviation for 50 episodes and 5 seeds.
>
> |  Tasks | BC | CC | DC |
> |  :------: | :------: | :------: | :------: |
> | Avg. R $\uparrow$ | 542.71 ± 9.63 | 369.76 ± 2.92 | 559.11 ± 46.05 |
> | Avg. CV $\downarrow$ | 0 ± 0 | 0.68 ± 0.46 | 0.57 ± 0.22 |
> | Avg. R-G $\uparrow$ | 545.50 ± 11.99 | 369.14 ± 2.16 | 545.85 ± 54.97 |
> | Avg. CV-G $\downarrow$ | 0 ± 0 | 0.78 ± 0.51 | 0.75 ± 0.28 |
>
> From the experiments, we can observe the following:
>
> **(1) The performance around the behavior policy conditions $[20, 30, 40]$ is stable and good in terms of both reward and constraint satisfaction.** Within the threshold condition range from $10$ to $50$, the versatile agent can adapt to unseen thresholds with reasonable task reward and near-zero cost violation.
>
> **(2) The selection of behavior policy conditions is crucial for the performance over the target condition interval.** The rewards concerning threshold conditions from $50$ to $70$ do not achieve the highest possible value. This sub-optimality may arise from challenges in estimating $z(\epsilon)$ in $Q(s,a | \epsilon) = \psi(s, a)^T z(\epsilon)$, when $\epsilon$ is too far away from the behavior policy conditions. In Theorem $1$, the bounded estimation error is derived based on the assumption that behavior policy conditions divide the target threshold condition interval equally. How to address the proposed problem is an interesting question and can be a significant area for future work. For example, we may introduce some additional regularizer into the $z(\epsilon)$ model to resolve this.
>
> We will add related discussions in the revision.
>
>
> **Reference:**
>
> [R-1] A. Ray, J. Achiam, and D. Amodei, “Benchmarking Safe Exploration in Deep Reinforcement Learning,” 2019.
>
> [R-2] Stooke, Adam, Joshua Achiam, and Pieter Abbeel. "Responsive safety in reinforcement learning by pid lagrangian methods." International Conference on Machine Learning. PMLR, 2020.

---

> > ### Comment · Reviewer_hfso · 2023-08-16
> > **Rebuttal response**
> >
> > I am satisfied with the authors' response. I will increase my score to 7.

---

> > > ### Author Response · Authors · 2023-08-17
> > > **Thank you!**
> > >
> > > We sincerely appreciate the reviewer's recognition of our paper and raising the score. Thanks again for the constructive comments!

---

### Official Review · Reviewer_R5zk · 2023-07-25

**Soundness:** 3 good
**Presentation:** 2 fair
**Contribution:** 3 good
**Rating:** 7
**Confidence:** 3

**Summary:**

The study investigated a versatile, safe reinforcement learning problem and proposed the Conditioned Constrained Policy Optimization (CCPO) algorithm. CCPO may be helpful in ensuring safety under the settings of various constraint limits. Moreover, the experiment results demonstrate that CCPO works well.

**Strengths:**

1. The problem is interesting.
2. The method may be useful for safe RL research.


**Weaknesses:**

1. The code the study provided is not convenient to run; we have to download the code file by file. Could you provide the whole code? Such that we can examine the algorithm better.
2. Please check the whole paper, and improve the writing quality, e.g., on page 3, line 101, please check the sentence, and revise “z(1)” as “(1)”.


**Questions:**

1. How can we handle non-convex optimization while ensuring safety?
2. If the cost limits are linear, what will happen?
3. Could you compare it with CPO-based methods?


**Limitations:**

1. The balance between reward and cost is not handled well.
2. Although the Lagrangian-based methods can not perform well on various cost-limit settings, their reward performance is better than CCPO.

---

> ### Author Rebuttal · Authors · 2023-08-08
>
> We thank the reviewer for your time and valuable feedback. We address the concerns as follows.
>
> > W1: The code the study provided is not convenient to run; we have to download the code file by file. Could you provide the whole code? Such that we can examine the algorithm better.
>
> We apologize if the code of the current version brings any inconvenience. We have submitted a new version to AC based on the review guideline.
>
>
> > W2: Please check the whole paper, and improve the writing quality, e.g., on page 3, line 101, please check the sentence, and revise “z(1)” as “(1)”.
>
> We sincerely appreciate your detailed review of our work. We will correct the typos and further improve the writing quality in the revision.
>
> > Q1: How can we handle non-convex optimization while ensuring safety?
>
> Thank you for your question. We decompose the general non-convex safe RL problems into two steps (E-M steps), where the first step handling safety constraints is a convex optimization problem. Here are the details: In the Constraint-Conditioned E-step, where safety is encoded, the policy optimization process is formulated as Eq. (8). We calculate the optimal variational distribution, denoted as $q(\cdot | \epsilon)$, by solving its dual form (10). Subsequently, we can obtain the closed-form solution for Eq. (8). In Appendix B.3, we demonstrate that the dual problem (10) is convex, and under certain mild conditions, we can ensure its strong convexity. Thus, various off-the-shelf convex optimization solvers can be utilized to find the optimal solution.
>
> > Q2: If the cost limits are linear, what will happen?
>
> Thank you for your question. In our experiments, all the behavior policy cost conditions exhibit a linear relationship with respect to the behavior policy index, indicating that the condition set forms an arithmetic progression.
>
> In terms of our Critics linear decomposition (starting from line 132): the linearity is describing the operational relationship between $\psi(s, a)$ and $z(\epsilon)$, while $\psi(s, a)$ and $z(\epsilon)$ are both nonlinear functions. We also provide some verification results for Q functions decomposition/estimation in Appendix D.1.
>
> We welcome any additional clarification or comments you may have on this problem.
>
> > Q3: Could you compare it with CPO-based methods?
>
> Thank you for your valuable feedback. We have made modifications to CPO, creating the Versatile CPO (via policy linear combination), and conducted experiments in all five tasks. The following table presents the comparison results between Versatile CPO and our proposed CCPO. It is evident that the proposed CCPO method outperforms Versatile CPO in terms of both constraint satisfaction and reward efficiency.
>
> | Tasks | stats | BC | CC | DC | DR | AR | Average |
> | :------: | :------: | :------: | :------: | :------: | :------: | :------: | :------: |
> |  Versatile CPO | Avg. R $\uparrow$ | 686.08 ± 2.58 | 455.87 ± 1.51 | 565.98 ± 5.71 | 304.09 ± 32.17 | 602.05 ± 1.66 | 522.81
> |  | Avg. CV $\downarrow$ | 1.77 ± 0.67 | 4.50 ± 2.08 | 6.32 ± 2.32 | 11.82 ± 5.85 | 2.19 ± 0.63 | 6.60
> |  | Avg. R-G $\uparrow$ | 676.27 ± 2.32 | 454.62 ± 1.97 | 559.71 ± 7.32 | 300.85 ± 32.41 | 592.89 ± 2.68 | 516.87
> |  | Avg. CV-G $\downarrow$ | 2.09 ± 0.59 | 4.94 ± 2.37 | 6.75 ± 2.47 | 13.07 ± 5.44 | 2.28 ± 0.73 | 5.83
> | CCPO | Avg. R $\uparrow$ | 710.86 ± 20.47 | 406.06 ± 6.30 | 630.55 ± 40.03 | 458.69 ± 12.98 | 660.88 ± 4.82 | **573.41**
> |  | Avg. CV $\downarrow$ | 0.59 ± 0.31 | 1.60 ± 0.91 | 0.32 ± 0.38 | 0.23 ± 0.25 | 3.13 ± 1.67 | **1.17**
> |  | Avg. R-G $\uparrow$ | 699.04 ± 20.48 | 401.53 ± 5.59 | 625.51 ± 40.12 | 455.64 ± 11.83 | 660.07 ± 5.26 | **568.36**
> |  | Avg. CV-G $\downarrow$ | 0.83 ± 0.42 | 1.49 ± 0.38 | 0.47 ± 0.55 | 0.33 ± 0.37 | 3.25 ± 1.48 | **1.27**
>
> > Limitation: The balance between reward and cost is not handled well; Although the Lagrangian-based methods can not perform well on various cost-limit settings, their reward performance is better than CCPO.
>
> Thank you for raising this point. It is true that the rewards of baselines in some tasks are slightly higher than our method's. However, we would like to gently point out that this is a common trade-off between task utility and safety performance in the safe RL literature -- the **optimal** policy under safety constraints might not be the most rewarding one[R-1, R-2, R-3].
> To be more concrete, the evaluation criteria in safe RL are:
> 1) method A is better than method B if A achieves better safety performance than B.
> 2) If both A and B satisfy constraints, the one with the higher reward is better.
>
> With the above metrics, we can conclude that CCPO outperforms baselines in most experiments. We will make the evaluation criteria clear in the revision to avoid confusion.
>
> **Reference**
>
> [R-1] Yu, Haonan, Wei Xu, and Haichao Zhang. "Towards safe reinforcement learning with a safety editor policy." Advances in Neural Information Processing Systems 35 (2022): 2608-2621.
>
> [R-2] Stooke, Adam, Joshua Achiam, and Pieter Abbeel. "Responsive safety in reinforcement learning by pid lagrangian methods." International Conference on Machine Learning. PMLR, 2020.
>
> [R-3] Liu, Zuxin, et al. "On the Robustness of Safe Reinforcement Learning under Observational Perturbations." The Eleventh International Conference on Learning Representations. 2022.

---

> > ### Comment · Reviewer_R5zk · 2023-08-14
> > **upgrade score**
> >
> > Thank you for your feedback. I am pleased with the response and intend to revise the score in favor of acceptance.

---

> > > ### Author Response · Authors · 2023-08-17
> > > **Thank you!**
> > >
> > > We sincerely appreciate the reviewer's recognition of our paper and raising the score. Thanks again for the constructive comments!

---

### Author Rebuttal · Authors · 2023-08-08

Dear reviewers,

We sincerely appreciate your valuable and constructive feedback on our paper. In addition to addressing each reviewer's comments, we'd like to highlight the new examples and experiments we added during the rebuttal phase. You can find the figures we reference in the attached PDF file.

### 1. Additional Illustration and Examples:
We are pleased to hear that you found our problem formulation interesting (R5zk, hsfo, 5QUs, TpqY), novel (5QUs), well-motivated (hsfo), and our method useful (R5zk), impressive and elegantly designed (5QUs). In response to your feedback, we have included more motivating examples for the problem settings, explanatory illustrations for the methodology, and a (shorter) algorithm diagram to provide clearer visual context.

### 2. New Experiments:
We are grateful for your positive feedback on our extensive (tMdc, 5QUs) and impressive experiments (5QUs). As a result of your valuable insights, we have incorporated the following new experiments to further enhance our work:

- (Reviewer R5zk, 5QUs) **CPO-Based Baseline**: To showcase the superiority of our approach, we modified CPO to create the versatile CPO and performed comprehensive experiments. The results demonstrate that CCPO consistently outperforms the versatile CPO, reinforcing the effectiveness of our method.
- (Reviewer hsfo) **Exploring More Behavior Policy Conditions**: In response to your suggestion, we conducted additional experiments with varied behavior policy conditions. These experiments highlight CCPO's ability to maintain safety for unseen threshold conditions, and we also offer further ideas to address this challenge.

We genuinely appreciate your time, attention, and valuable feedback.

Best regards,

The Authors

---

### Decision · Program_Chairs · 2023-09-21

**Decision:**

Accept (poster)

**Comment:**

The authors provided extra code. The reviewers reach a consensus that this paper should be accepted. It would be better to improve the experimental part by comparing to existing Safe Policy Optimization methods.